# On Time, Leisure, and Health in Retirement: Implications for Public Health Services

**DOI:** 10.3390/ijerph20032490

**Published:** 2023-01-30

**Authors:** Susan Hutchinson, Douglas Kleiber

**Affiliations:** 1School of Health and Human Performance, Dalhousie University, Halifax, NS B3H 4R2, Canada; 2College of Education, University of Georgia, Athens, GA 30602, USA

**Keywords:** leisure, leisure education, life management strategies, retirement, well-being

## Abstract

Various life challenges, such as widowhood, poor health, or significant caregiving responsibilities, can make the possibility of how to spend one’s time in retirement seem daunting. Planning can help people feel more confident and prepared. In this paper, we review research that has examined: (1) life factors impacting fears about and adjustment to retirement, (2) access to resources and utilization of strategies that impact adaptation processes, and (3) the ways leisure and leisure education may be resources to support not only individual adaptation but practices of public health service providers in assisting people who may be struggling with this transition. The review ends with recommendations for public health practice including: (1) the inclusion of leisure and leisure education as a focus of service provision; (2) the development of partnerships or collaborations between public health and recreation-related organizations; and (3) the development and delivery of group- and individual-based leisure education programs.

## 1. On Time, Leisure, and Health in Retirement: Implications for Public Health Services

One’s relationship with time is both personal and shaped by deeply held social norms about the good and right way to use it [1]. In fact, one of the most ubiquitous statements about time in later adulthood is about how precious it is; people cannot buy more time, so they should not waste what is available. Yet, in the context of planning for the transition to retirement, what does the prospect of more time mean for someone who has just lost a life partner, for example? Or for someone whose physical abilities have been irrevocably altered because of disease or illness? In these circumstances, time implies the possibility of a future of pain and suffering, with limited respite (at least initially) from one’s own thoughts, no less opportunities for self-expression.

Of course, these are the extremes of human suffering, and problems with time (both time scarcity and abundance) in retirement seem more understandable in such circumstances. However, time is also troubled in situations that are less tragic but difficult, nonetheless. For example, health issues may precipitate the decision to retire, or individuals may feeling ‘forced’ out of work due to changes in their organization’s practices or policies (e.g., [2,3]). In these cases, aspects of one’s job may still hold valued meanings, even if the work context does not. Alternatively, people who are financially able to retire may elect to remain at work to distance themselves from having to deal with even greater life challenges outside of work.

While many people look forward to retirement and welcome its arrival, clearly, there are others who do not (e.g., [4]); in fact, for them, continuing work provides a safe haven against the anticipated vagrancies of life that await them in retirement. Although financial concerns and plans are often paramount in people’s thinking about the transition to retirement (e.g., [5]), in this review, we focus on life circumstances that make the transition to retirement challenging. In such circumstances, time management for whatever freedom retirement might afford may be all the more important.

The purpose of this review is to consider more deeply the life challenges impacting some individuals’ fears about and adjustment to retirement and to examine the ways access to resources and utilization of strategies can impact adaptation processes. We also consider the ways leisure and leisure education may be resources to support not only individual adaptation but practices of public health service providers in assisting people who may be struggling with this transition. In doing so, we make a case that: (1) time is a precious commodity in later life made even more so by challenging post-retirement circumstances; (2) the health and well-being of older at-risk populations depend to a great extent on optimizing limited time and other resources available through the use of life management strategies; and (3) finally, leisure education is a psycho-educational mechanism that can be utilized within the public health sector to support people whose health and well-being is impacted by challenging life circumstances in the transition to retirement.

## 2. Retirement and Well-Being

In general, for the majority of people, the transition to retirement is associated with improvements in mental and physical health, including improved mood, perceived well-being, better sleep, and more energy, especially when retirement is associated with a sense of freedom from work-related stressors or time constraints [4,6,7,8,9]. Most people look forward to retirement to explore other facets of life and to spend more time with family and friends [9,10,11,12]. Yet, when retirement is perceived as forced or unwanted, it is often associated with a significant reduction in well-being and mental health, including increased depression, stress, loneliness or boredom, excessive alcohol use, or some combination of those (e.g., [4,13,14,15]). These outcomes are complex and ambiguous. As Olds and colleagues noted: “Retirement could also precipitate positive *or negative* changes in identity, social interaction and intellectual activities, as well as in financial security and family relationships, all of which are known to impact on mental health” [4] (p. 2; italics added). 

Declining or poor health and relationship status does seem to negatively impact the adjustment to retirement. For example, Wang and associates examined predictors of retirement adjustment and found that physical health decline, marital status (e.g., becoming widowed), attachment to work, and forced retirement were among the factors associated with poorer adjustment (which is also to say that being happily married, in reasonably good health, with sufficient income for basic needs, and engaged in meaningful work, even if it is unpaid, are protective factors) [3]. In their systematic reviews, both Barbosa and colleagues and Amorin and Franca found physical health to have the greatest impact on retirement adjustment followed by finances, exit conditions (e.g., being forced to retire early), marital status, and interpersonal relationships [2,16]. However, Barbosa and colleagues also suggested that leisure and social integration and retirement preparations were additional important factors not only for successful adjustment but also for retirement well-being. A recent meta-analysis by La Rue and colleagues, examining over five decades of studies regarding factors predicting psychological adjustment to retirement, identified social participation and physical health as the most important factors to retirement adjustment [17]. Of the subfactors associated with social participation (e.g., frequency of social interaction, etc.), ease of maintaining social ties had the greatest influence on retirement adjustment, suggesting “that it is the variation in people’s ability to overcome their concerns about being socially excluded or losing existing social ties that is particularly important” [17] (p. 11). This seems particularly relevant for those who are contemplating a single life in retirement.

With respect to the impact of caregiving on retirement adjustment, more research is needed, but an earlier study by Detinger and Clarkberg found that women were five times more likely to retire to care for a family member, with nearly two-thirds of women nearing retirement providing informal caregiving [18]. Although gender roles may be shifting as the baby boomer generation ages and more women are remaining longer in the workforce, it does seem that caregiving responsibilities continue to be a time management issue for women and men who feel obligated to take on even more caregiving responsibilities if/when they retire. While the Detinger and Clarkberg study is now two decades old, there are no new indications that these statistics have changed.

These findings highlight the importance of health and social resources in preparing for the transition to retirement. However, surprisingly, they say little about how people adapt to or manage these life challenges and the time available as they negotiate the retirement transition. In the next sections, we consider the resources people have access to in preparing for retirement, including leisure.

## 3. Access to Resources in the Transition to Retirement

All the factors identified above that impact the adjustment to retirement—social networks, marital status, health status, opportunities for self-expression, living space, emotional strength, and time itself—can also be considered resources that people have to draw on as they prepare for and negotiate that transition [19]. Resources, in general terms, can be considered a stock or supply that individuals can access to function effectively. For example, someone who is retiring with a modest pension, without mobility limitations, volunteer commitments to look forward to, and a strong network of friends outside of work can be considered to have good access to a range of resources. This is contrasted with someone who may need to continue some form of work to cover living expenses, lives with chronic pain that restricts physical activities and reduces mood, and has few social connections or interests outside of work. Overall, the greater people’s stock of personal, social, and health resources, the more likely they are to fare well in the transition to retirement, even when living alone or with health challenges [3,13].

According to Wang and colleagues’ ‘resource-based dynamic’ model of retirement, adjustment to retirement depends on available physical/cognitive, motivational, financial, social, and emotional resources, with high resource availability allowing faster and better adjustment to retirement [3]. Principi and associates concluded that psychological resources (e.g., optimism and determination) are especially important for retirement satisfaction, regardless of whether or not people have retirement plans [20]. Other researchers have noted the importance of having a sense of mastery (or confidence to take action on priorities; [21,22]) and sense of purpose [23,24] as other important psychological resources for retirement adjustment, perhaps even more important than health, income, housing, or relationship status.

In summary, it is clearly important to help people examine the personal, social, and lifestyle resources they bring with them into retirement. A part of this is leisure; it is a lifestyle resource that dramatically impacts health and well-being and is often central to people’s goals for retirement.

## 4. Leisure as a Resource for Managing Life Challenges and Optimizing Well-Being through the Retirement Transition

The common view of leisure is that while it may be associated with well-being and happiness—particularly in later life—it is most often considered ‘self-indulgence’ and idleness and may, even in retirement, be regarded as wasting time [1]. We and other leisure scholars [25] take a contrary view, believing that leisure not only offers the potential for contributing to self-actualization and social integration, but it may also have functional value in enabling one to cope with difficult circumstances, including those that are compromising to both mental and physical health [26,27].

In contemporary leisure studies, leisure is often defined as freely chosen, personally meaningful, and enjoyable activities or experiences [25]. The idea that leisure is not the same as idleness or even ‘free time’ dates back to ancient Greece. In his classic *Of Time, Work and Leisure*, Sebastian deGrazia noted that Aristotle’s view of leisure, or *schole*, was that it was a condition of peace and quiet, in some opposition to action, and with time being irrelevant [28]. In other words, leisure was originally seen to be that ideal state available to human beings where peace was afforded through civilization. Over centuries, with industrialization and modernity, leisure became generally associated with work, as time was left over for recuperation or the activity of re-creation itself, making a return to productive labor more efficient and effective (e.g., [29]). In addition, for those with wealth and other circumstances allowing freedom from the necessity of work, it became a symbol of materialist success and evoked a variety of responses from contempt to admiration (cf. Linder, 1970; Veblen, 2014 (1899)) [30,31]. However, leisure studies scholars, while recognizing this evolution and critical analysis, have insisted that leisure is a subjectively-experienced context where activities are carried out for their intrinsic value (or not at all), independent of any relationship to work [25].

Even in the absence of a relationship with work or social status, the experience of self-determination and opportunity for intrinsically motivated action do have functional value for development across the lifespan [32], for promoting health and well-being [33,34], and for adapting to stressful and challenging life circumstances such as living with chronic health conditions [35,36,37] or the death of a spouse (e.g., [38,39]). For example, for the participants living with chronic kidney failure in McQuoid’s study, leisure served as a resource for both coping and managing their illness long-term, providing opportunities to experience greater control and a sense of normalcy [36]. Standridge and colleagues found that, by seeking out new social groups, leisure provided recently widowed women with opportunities for distraction, companionship, and support, all of which enabled the reconstruction of their ‘new reality’ (as a single person) and sense of self [39].

In the context of challenging life circumstances, people’s access to or availability of personal and social resources *and* leisure can be diminished; however brief moments of leisure (i.e., casual leisure) can still be accessible, meaningful, and beneficial [40,41,42,43]. Casual leisure can range from playing cards or connecting through online communities [42] to leisurely walks. Regardless of the form it takes (e.g., physical or creative, individual or with others), there is evidence that casual leisure can: (1) serve as a source of positive distraction buffering immediate stress and sustaining coping efforts (having something to look forward to); (2) preserve or restore one’s sense of self (e.g., affirming values); and (3) contribute to growth-oriented change [40]. Other research has identified the importance of both flow-like ‘serious’ leisure activities (e.g., those that require a significant investment of skills and attention [43,44,45,46,47], and more meditative aspects of leisure that cultivate mindfulness and savoring [48,49]). Personally meaningful leisure experiences—those that allow for authentic expression of oneself—are also highly valued [50]. All these forms of leisure are worth identifying when considering how to best support people planning their time use in the context of challenging life circumstances.

Since life challenges inevitably relate to decisions about time use, attitudes toward time and time use may be central to understanding its functional value. Furthermore, knowledge and awareness of strategies for managing challenges in order to make the most of one’s time are equally important. In the next section, we consider the strategies that have been identified for managing time and life and how they might apply to the retirement transition when other resources are minimal or compromised.

## 5. Time and Life Management Strategies

Beyond the health, personal or social resources identified previously, it is important that people have developed a range of adaptive strategies to allow them to anticipate and manage challenges, including those that are time-related, that they will encounter in the retirement transition [17]. Interestingly, the management of time has most often been associated with time scarcity; in other words, making the most of the limited time one has. Yet, if nearly every hour of the day is ‘free time,’ how can this time be structured to avoid boredom or a sense of uselessness? Alternatively, if one anticipates endless responsibilities (e.g., for caregiving), how can people manage their time to ensure they save some time to care for themselves?

Although the concept of time management is clearly relevant in relation to the retirement transition, it does not fully address the adjustments needed to experience one’s time as meaningful or health-enhancing when faced with life circumstances that make time use challenging. Managing most life challenges inevitably involves deciding where to invest one’s time and then drawing on available social, personal, or health resources to make it happen. Furthermore, planning for time use when encountering life challenges is less about scheduling and more about planning and problem-solving, though the actual actions do require some intentional investment of time.

In addition to personal resources such as motivation or optimism that each of us possess in varying degrees, access to a set of life management strategies may be particularly important. According to Jopp and Smith, life management strategies are learned and practiced skills [51]. Importantly, whether directed towards managing health concerns or other life challenges, strategies are the skills—enacted through thoughts and behaviors—that people utilize to actively optimize remaining resources and manage challenging life circumstances. In the context of caregiving responsibilities, for example, strategies might include planning for respite care to get out of the house or meaningful at-home activities. Jopp and Smith found that life management strategies “serve as protective buffers by balancing the effect of low personal resources on well-being” [51] (p. 262) but only when people had confidence in the use of these strategies. Looking beyond personal and social resources such as health, social network, income, or education, Jopp and Smith suggest that self-regulatory processes (e.g., goal adjustment, control beliefs, and life management strategies) significantly contribute to what and how people invest their time and other energies or utilize compensatory strategies to optimize both engagement and positive outcomes [51]. Notably, the concept of life management strategies overlaps substantially with the idea of chronic condition self-management (CCSM) in the health/public health context. According to Barlow and colleagues, chronic condition self-management refers to “individuals’ ability to manage the symptoms, treatment, physical, and psychosocial consequences and lifestyle changes inherent in living with a chronic condition” [52] (p. 178).

Leisure researchers have identified diverse strategies that individuals utilize when they encounter constraints to leisure participation that seem consistent with the concept of life management strategies. Many are either attitudinal or behavioral and all relate to selecting, optimizing, or compensating for limitations related to leisure and time use in order to continue to experience the beneficial effects of leisure for health and well-being [53,54,55,56]. For example, in a study by Hutchinson and Nimrod, participants living with chronic conditions identified personal (e.g., positive attitude and prior leisure experiences) and social (e.g., friends, family, and work colleagues) resources as well as a number of internal and behavioral strategies to take action on leisure-related goals, such as: committing to leisure for health purposes; using cognitive (e.g., self-talk) or behavioral (e.g., using compensatory aids) strategies to continue to participate in valued activities, even if in modified ways; and setting leisure-based goals that met personal needs (both for maintaining and enhancing health and getting more out of life) [53]. These same strategies seem relevant in the context of planning for and managing one’s time in retirement when faced with challenging life circumstances.

Other studies have examined strategies used by retirees to manage time and other life challenges. For example, despite being relatively healthy, most retirees in Kleiber and Nimrod’s study reported three to four constraints to their time use, including physical limitations (e.g., due to injury or surgery), caregiving responsibilities (e.g., for spouse or grandchildren), financial, personal (e.g., indecision and anxiety), residential relocation, and interpersonal factors (e.g., lack of companion and spouse’s preferences) [56]. In addition to reporting feeling of frustration, some reported feeling acceptance and gratitude in the face of these constraints. Participants also described a number of strategies they employed to overcome constraints to using their time in ways that mattered to them, including: (a) eliminating some activities and pairing down others (especially those that were less satisfying, or more difficult to carry out) leaving more time for what was really important; (b) persisting and in some cases deepening commitments to activities that were especially important; (c) substituting: while goals may stay the same, substituting one activity for another was a common strategy to respond to constraints; and (d) exploring and self-discovery wherein some participants viewed constraints as an ‘opportunity’ to experiment with new possibilities for their lives [56].

Genoe and colleagues examined phases of the transition to retirement among baby boomers in Canada and also found their participants utilized a number of strategies to manage changes in their time use [57]. While some described taking on part-time, contract work, or bridge employment in preparation for the transition, others described having different priorities for and attitudes towards their time use in the initial stages of retirement, including valuing downtime and opportunities for spontaneous activity as well as avoiding activities requiring long-term time commitments (e.g., taking on volunteering). It was primarily after the initial ‘honeymoon’ phase that the study participants enacted more specific strategies for time use including creating structures that allowed for both flexibility and structure but that also ensured life priorities (e.g., related to health and social connections) were addressed.

In summary, it is important that people facing what are perceived as overwhelming life challenges in anticipation of retirement are supported in developing knowledge, skills, and confidence to manage their time in retirement, including learning strategies to address constraints to personally meaningful time use. The strategies described above are applicable to managing both life challenges and time in the transition to retirement. From our perspective, leisure education is a key solution and resource for public health service providers to assist people who do not possess these time management skills and adaptive strategies.

## 6. Implications for Public Health Practice: Leisure Education

Retirement will require ongoing adjustment, even in the best cases, and those adjustments will have to do with time allocation and adjustment to an awareness of a narrowing of time left to live. While many people will have figured this out, others may not; and still, others are trying but failing. As Rosenketter and colleagues noted:

Health care professionals need to be constantly aware that for at least some individuals, retirement can be a negative influence—both for retirees and their families. A potential lack of adequate adjustment suggests the need for an assessment of the impact of retirement on a person’s health status and for therapeutic interventions to facilitate the adjustment process. [58] (p. 16).

They further suggest that, as part of this, there is “a need for early planning and programs on the use of time and management designed to meet individualized needs in preparation for retirement and during postretirement adjustment” [58] (p. 1). Leandro-Franca and colleagues argued that older adults are at risk of negative health outcomes if proactive steps are not taken to age well; retirement programs that actively promote well-being can reduce this risk [59]. “Balanced against the alternative—that is, a cohort of poorly prepared retirees who can be expected to experience a diminished quality of life—it would seem well worth the cost to develop evidence-based programs that have clearly focused objectives and demonstrable effects” [59] (p. 510). As it relates to the life management strategies described earlier, Jopp and Smith also suggested that “enhancing the use of […] life-management strategies by instruction and training might be promising for interventions focusing on dealing with age-related change” [51] (p. 262).

Leandro-Franca and colleagues suggested that both long (8–20 weekly sessions) and brief (e.g., one–three sessions and group workshop format) retirement planning interventions can promote cognitive (e.g., decision-making and knowledge acquisition), motivational (e.g., clarifying retirement goals), and behavioral (e.g., steps that can be taken to strengthen one’s social support network or daily health practices) changes [59]. Leisure education is one highly relevant and accessible psycho-educational modality that can effectively address these aims.

Leisure education applies theory and evidence about leisure and leisure behavior to the design of experiential or learning-oriented activities intended to enhance participants’ knowledge, skills, awareness, and confidence related to leisure and recreation participation [60,61]. For example, depending on the learning needs of individuals or groups, Dattilo recommends that leisure education focuses on helping people develop: a leisure ethic (e.g., value leisure), awareness of themselves and leisure, awareness of leisure-related resources or opportunities, and skills needed for various leisure activity pursuits as well as skills to build social connections, make decisions and manage challenges [60]. While the ultimate goal of leisure education is to help people do more of what matters to them in ways that contribute to their overall health and well-being, a key focus of leisure education could be to help people develop ‘life management’ strategies that can be used to effectively manage time and other challenges in the transition to retirement. As noted earlier, these strategies are often only effective when people have confidence in their abilities to enact them [51]; further confidence or a sense of mastery is likewise considered a key personal resource for retirement [21,22].

Carbonneau and colleagues heard from recent retirees that they had three main learning-related needs: to develop more positive attitudes toward retirement and the role of leisure, to better understand their retirement needs, and to develop skills and knowledge about leisure resources [62]. Kleiber and Linde reviewed a wide range of retirement and leisure-related evidence to suggest that leisure education address two key goals to support successful adjustment in retirement: (1) consider what is lost in the transition to retirement (e.g., opportunities to express competence) and how leisure can serve as a context for meeting these enduring needs (i.e., to compensate for or replace what is most valued about work); and (2) focus on examining the various ways different forms of leisure can be a ‘restorative resource’ for health and well-being, including a focus on addressing social needs [63]. To address both goals, Kleiber and Linde suggested that retirement planning programs focus on helping people: (a) deepen self-awareness and clarify values, (b) identify resources or opportunities (e.g., within one’s neighborhood or community), (c) have the chance to ‘experience’ leisure (e.g., leisure sampling), and (d) engage in problem solving that supports making decisions and commitments [64]. The first recommendation is particularly important for people whose attitudes toward time use and leisure may serve as barriers in the transition to retirement and the latter is essential for people experiencing challenges identified earlier. These suggested topics are consistent with the learning needs identified by Carbonneau and colleagues [63]; furthermore, both align with Dattilo’s leisure education content areas identified above (e.g., developing a leisure ethic and awareness of self and leisure resources; developing activity and decision-making [problem-solving] skills) [60]. An important focus of Dattilo’s leisure education model is ‘managing challenges’ [60]; while neither Carbonneau et al. nor Kleiber and Linde identify this within their recommendations, it does seem to be an important missing focus for the development of leisure education interventions explicitly targeting people struggling with the transition to retirement [60,62,63].

Recently Woodford and colleagues applied Dattilo’s and Kleiber and Linde’s ideas to the development of a brief (six sessions) retirement lifestyle planning program delivered in a university context [60,63,64]. In the context of the program, participants were encouraged to identify personal, social, and health-related resources needed to live a fulfilling life in retirement; to assess their personal strengths; to consider time use (including the importance of finding purpose and the ways leisure/recreation can be resources for living a fulfilling life in retirement); and to identify potential barriers and strategies to taking action on life priorities (now and in retirement). Each of these program goal areas are relevant for helping people develop knowledge, skills (e.g., life management strategies), and attitudes (including awareness and confidence) for time use in retirement. Program participants reported feeling both more confident and prepared following the program [64]. Although their program addresses general lifestyle planning needs in the retirement lifestyle planning process, there was limited attention paid to the unique challenges people may be experiencing. There is a need for future program development that specifically addresses challenges related to time use, such as how to navigate caregiving responsibilities, being widowed or single, and health limitations in the transition to retirement. One option for this additional programming might be to offer specialized workshops following a general retirement program (which would allow people to begin to identify their own needs for further education) or to partner with service organizations (caregivers’, bereavement/widows/widowers’, or cancer/arthritis self-help or support groups as examples).

As it relates to managing health challenges, Hutchinson and colleagues recommended the incorporation of leisure education within the health and medical services that are concerned with chronic condition self-management (CCSM), and they provided examples of existing interventions as well as suggestions for topics or content of leisure education programs [65]. They also strongly advocated for including leisure education “within the rehabilitation and healthcare sector to ensure leisure is recognized as a legitimate and valuable aspect of CCSM and that infrastructure and systems are in place to optimize leisure opportunities” [65] (p. 3). Their recommendations seem applicable to supporting individuals struggling with the transition to retirement, especially those for whom retirement is problematic and impacts health and well-being.

Specifically, when thinking about the public health services context and based on the foregoing review, we recommend:*The inclusion of leisure and leisure-education-based retirement planning as a focus of service provision*. To do this means that public health service providers need to become informed and educated about the benefits of leisure for health and well-being, and about the psychosocial and contextual factors impacting health and well-being in the transition to retirement. Furthermore, there is a need to also facilitate education amongst potential community partners as part of partnership building (see next recommendation).*The development of partnerships or collaborations between public health, health, social service, employers, and recreation-related organizations*. Partnerships between public health and recreation-related organizations (e.g., community centers and leisure service agencies) and between public health and other health and social service organizations, including service providers within workplaces, could create supportive environments that ease or bridge the transition between health-related and community-based services [66]. For example, retirement planning programs could be co-hosted within recreation or senior -serving centers, with opportunities to explore local community resources provided (or encouraged) in the context of the program delivery. Likewise, partnerships between public health service providers and a large employer or union group could lead to the co-hosting of retirement programs or referrals to specialized programs (as described above) after a more general retirement program has been offered.*The development and delivery of group- and individual-based leisure education focused on planning for time use in retirement*. The topics proposed by Carbonneau and colleagues and Kleiber and Linde and incorporated within the program developed by Woodford and colleagues are examples of content that could meaningfully address participants’ needs to develop greater self-awareness and knowledge as well as to learn and practice strategies to overcome constraints impacting both time use and health and well-being in preparing for the transition to retirement [62,63,64]. As suggested above, there would also be merit in developing specialized workshops targeting specific health or wellness challenges (e.g., caregiving, widowhood, and health limitations) and, for those with unique or complex needs, it would be helpful if they could be referred to wellness *navigators* or *life coaches* who would be ideally positioned to support people whose needs cannot be addressed in a group education context [67]. A final recommendation is to include those with similar lived experiences (e.g., recently retired, living with health limitations, and caregiving) as co-facilitators within a group education program in order that they can share their experiences and stories with program participants.

## 7. Conclusions

As identified in this review, research on satisfaction with retirement suggests that most people are generally satisfied and happy to have retired, particularly those who have maintained a pattern of activity (leisure- or voluntary-work-related) that affords satisfying engagement and some sense of purpose [58]. However, those people are generally well-resourced (e.g., in terms of health or finances) and relatively untroubled by other life circumstances. Clearly, however, there is a substantial minority who find retirement stressful for the lack of those things and other reasons, including attitudes toward time and leisure. To the extent that people are not prepared for this transition, they are at risk of poorer health and well-being outcomes as they move forward into later life.

Effectively coping with and managing life challenges in the transition to retirement depends both on skills with time management and on the use of both internal and behavioral strategies to meet personal needs and goals [22,51]. In addition to other personal resources (e.g., confidence and optimism), in this review, we argued that leisure is itself a relatively accessible health resource. But people’s views about and experiences with leisure and time use can impact the extent to which they access leisure to not only manage challenging life circumstances in the moment but to plan for time use and activities that will be personally meaningful and health-enhancing in retirement. However, future research is needed to determine if these propositions hold true and to better understand the strategies effectively employed to manage time use and other challenges specifically in the context of the transition to retirement.

While many people may naturally possess and utilize a variety of strategies to manage challenging life circumstances, there are those who do not. To address this unmet need, we argued for the inclusion of leisure and leisure education in the delivery of public health and community programs and services to support people who are experiencing deleterious challenges in managing life and time. Creating partnerships between public health, health, employers, and community recreation/social services focusing on the delivery of leisure education programs and wellness navigator services seems to be a cost-effective solution to promote health and reduce the burden on primary health care systems. To this end, in their conclusion to their qualitative analysis of retirement planning programs, Leandro-Franco and colleagues noted:

“From a public policy perspective … it would be well worth investing in the further development and dissemination of retirement intervention programs. This endorsement is conditioned upon the premise that future program specialists would consider best practices criteria when designing their research and when developing program content … It could be advantageous for proponents of interventions to partner with universities, large corporations, and government agencies when seeking funding and tangible support for their efforts” [59] (p. 510).

Despite this endorsement of the need for more retirement intervention programs, Leandro-Franca noted that there is a continuing need for a more robust evaluation of them [59]. Relatedly, while there is some evidence of the effectiveness of leisure education interventions to address the needs of people living with ongoing health problems and more generally in preparing for the transition to retirement [64,68,69,70], there is a need for the continued development and evaluation of innovative individual or group leisure education programs and services that address the complex needs of some individuals facing this transition.

A final key unanswered question is, how do public health providers reach people who may be struggling alone—and outside health or social service systems—with time use and other life challenges following retirement or who, because of living with ongoing challenges, have limited energy to think about developing time use skills or participating in an education program? We speculate that brief leisure education programs are likely beneficial in this context [69,70] along with tapping into existing systems of support (e.g., employee assistance programs and peer support programs). However, future research is needed to determine: (a) how people who are retiring find out about and coalesce necessary resources to attend, (b) the supports needed to effectively overcome challenges to access programs or services, and (c) the methods of program delivery likely to be most beneficial (e.g., in-person or online). Again, we suggest that here is where peer-support or peer-assisted learning models might be most accessible and beneficial; however, this proposition remains untested. Finally, there is evidence of the effectiveness of virtual education in relation to chronic condition self-management for older adults [71], and further research is needed to determine if programs focusing on challenges in the context of retirement could be equally beneficial.

## Data Availability

Not applicable: Existing, previously published manuscripts were used for this review.

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
