# Peer review of "On Time, Leisure, and Health in Retirement: Implications for Public Health Services"

_ijerph, 2023, doi:10.3390/ijerph20032490_

Round 1
Reviewer 1 Report
Thank you for the opportunity to review this manuscript. Overall, I found it to be well-written and engaging. It provides a clear and concise review of the current research regarding leisure time and the transition to retirement. I believe it makes a valuable contribution to the leisure and aging literature. I have a few minor revisions that the authors could consider in revising the manuscript.
· On page 1, line 27, consider revising “her life partner” to “their life partner”.
· Where applicable, it would be useful to include some references to support the arguments in the introduction
· The first sentence on page 2 is quite lengthy and could be broken into two sentences. For example, a period could be placed after processes, and a second sentence could begin with We also aim to review ways leisure…
· On page 3, it would be helpful to define cognitive reserve for the reader, who may not be familiar with the term
· On page 3, in the first paragraph in the section on Access to Resources, I suggest including an example to demonstrate how these resources may help people to transition to retirement. More specific examples would be beneficial throughout the manuscript to better illustrate the concepts and their implications for transitioning to retirement.
· It could be helpful to include more information about leisure education. Datillo provides a variety of components of leisure education – a brief description of these components and how they might be relevant to retirement transitions would be beneficial, particularly as they relate to Carbonneau et al.’s research findings and Kleiber and Linde’s recommendations for leisure education.
· I appreciate that the review includes several implications and considerations for practice. However, the review lacks suggestions for future research. Given the extent of the review, I’d appreciate understanding the authors’ perspectives on the limitations of the current body of knowledge and how future research might address these limitations.
Author Response
Please see the first table in the attachment. Thank you for your thoughtful review.

Reviewer 2 Report
Referee report on ”On Time, Leisure and Health in Retirement: Implications for Public Health Services”
Overall impression: The literature review is well written and broad. I only have a couple of comments.
My first main comment is that there are quite a lot of self-citations. I think in case of self-citations, the authors should specify in more detail how their own work specifically relates/contributes to the issue in question.
For example line 159- “Even in the absence of a relationship with work or social status, the experience of self-determination and opportunity for intrinsically motivated action do have functional value for development across the lifespan, for promoting health and wellbeing, and for adapting to stressful and challenging life circumstances (Kleiber, 1999; Walker et al., 2019) such as living with chronic health conditions, acquired disabilities, divorce or the death of a spouse (e.g., Hutchinson & Kleiber, 2005b; Hutchinson & Warner, 2014; Janke et al, 2008a,b; Kleiber et al., 2002; Kleiber & Hutchinson, 2010; Payne et al., 2006; Standridge et 165 al., 2020a).”
The second main comment is:
You make many suggestions regarding leisure education and advocate leisure education strongly. However, you don’t discuss at all how can one ensure that the persons who need this kind of education and persons with different challenges find these programs and participate? Those with challenges may have so many worries that they have no energy to think about developing their time use skills or to participate to some education programs. Is there any research about how people who have participated in some education programs have become aware that this kind of program exists? There is no benefit in offering leisure education programs if people don’t find them. This needs to be discussed, perhaps you could add a small subsection about this issue.
A couple of smaller comments:
Line 347: “There is a need for future program development that specifically addresses challenges related to time use, such as how to navigate caregiving responsibilities, being widowed or single, and health limitations in the transition to retirement.” Do you mean that this kind of topics would be part of more general retirement lifestyle planning programs, or do you mean that completely separate programs would be targeted to those facing challenges? This needs to be clarified.
Line 357: “…aspect of CCSM….” What is CCSM?
Author Response
Please see the attached file with responses to both Reviewers. The second table includes our response to your specific feedback and suggestions. We really appreciated the thoughtful suggestions you provided.
